# 16.8/15.2 ppm/°C 81 nW High PSRR Dual-Output Voltage Reference for Portable Biomedical Application

**Hongwei Yue, Xiaofei Sun, Junxin Liu, Weilin Xu \*, Haiou Li, Baolin Wei, Taotao Wang and Siyu Lin**

Guangxi Key Laboratory of Precision Navigation Technology and Application, Guilin University of Electronic Technology, Guilin 541004, China; guetyhw@163.com (H.Y.); sunxfsuperior@163.com (X.S.); liujxsuperior@163.com (J.L.); Lihaiou@guet.edu.cn (H.L.); guilinwxb@163.com (B.W.), wtao_1994@163.com (T.W.); guiguiyu123@163.com (S.L.)

**\*** Correspondence: xwl@guet.edu.cn; Tel.: +86-1397-830-2862

**Abstract:** A dual-output voltage reference circuit with two reference voltages of 281 mV (*Vref1*) and 320.5 mV (*Vref2*) is presented in this paper. With a novel and precise circuit structure, the proposed circuit, operating in the subthreshold region, integrates two different output voltages into a circuit to form a dual-output voltage reference, and cascode current mirrors are used to enhance the power supply rejection ratio (PSRR). The proposed circuit was designed in a standard 0.18-μm CMOS process and has a series of attractive features: low-temperature coefficient (TC), high-PSRR, low-Line sensitivity (LS), small-chip area and low-power consumption. Monte Carlo simulations for 2000 samples showed that the output voltages 281 mV and 320.5 mV had a variation coefficient of 1.73% and 1.44%, respectively. The minimum power consumption was 84.1 nW at 0.9 V supply, proving that the circuit is suitable for portable biomedical application. The active area of the proposed voltage reference was only 0.0086 mm$^2$.

**Keywords:** portable biomedical; dual-output voltage reference; low-Temperature coefficient; high-power supply rejection ratio; low-Line sensitivity; low-power consumption; Monte Carlo simulation; Integrated circuits (Analog)

## 1. Introduction

With the development of human body area networks such as wearable medical devices and medical measurements, a serious challenge is put forward to low-power and high-performance ICs [1]. As an essential module in analog and digital circuit systems, multiple output voltage references have been the most important part of low-power and high-performance ICs, and have been attracting increasing attention.

In recent years, even though several voltage references have been developed, traditional voltage references have many problems. In the literature [2], the traditional bandgap reference, which adopts BJTs, can generate stable voltage. However, this kind of circuit consumes too much area and power, and only has one output voltage, which cannot satisfy the development of ultra-low power and high-performance applications. In the literature [3], multiple circuits are used to generate multiple output voltages, which consists of two different start-up circuits, two different current reference circuits, and two different voltage reference circuits. Even though this technique can achieve dual-output voltage reference, this technique has many limitations, such as the structure of the circuit is complex, and the chip area and power consumption are larger. In the literature [4], the characteristics of subthreshold MOSFETs is used to build a low power voltage reference without resistors. Although

this effectively reduces the power consumption and temperature coefficient, the PSRR is not ideal. In the literature [5], a method to improve the PSRR by using the structure of feedback loop is presented, but the TC is not ideal, this technique cannot achieve dual-output voltage reference, and the overall power consumption of the circuit is larger. None of these circuits [1–5] meet the requirements for low power, low voltage, high precision, low area, and compatibility of CMOS process.

To overcome the existing mentioned problem, and to improve the overall performance of the multiple voltage reference, a novel and precise dual-output voltage reference circuit with low power consumption, low TC and high PSRR is proposed in this paper. The proposed circuit architecture consists of one start-up circuit, one current reference circuit, and two different simpler voltage reference circuits. The proposed voltage reference circuit is smaller and simpler.

## 2. Circuit Design

The proposed dual-output voltage reference circuit is illustrated in Figure 1. It consists of two parts, namely a start-up circuit and a core dual-output voltage reference circuit, and transistors of dual-output voltage reference circuit are operating in subthreshold region, except transistors of start-up circuit.

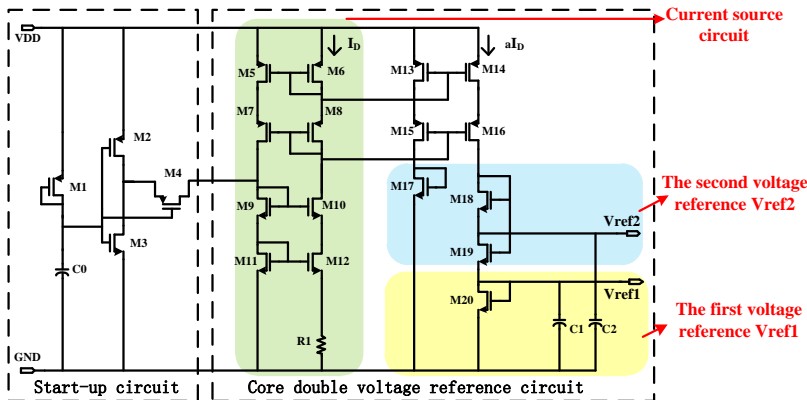

**Figure 1.** Proposed dual-output voltage reference circuit.

### 2.1. Start-up Circuit

The start-up circuit is composed of M1, M2, M3, M4 and capacitor $C_0$. When the system works normally, it will turn-off the start-up circuit and can reduce power consumption. The start-up circuit is used to obtain start-up current and get rid of the degenerate bias points.

### 2.2. Current Source Circuit

The Current source circuit is used to provide bias current for dual-output voltage reference circuit. M5 and M6, operating in the subthreshold region with the same aspect ratio, is a cascode current mirror. The bias current of the current source circuit is copied to the core dual voltage reference circuit through the cascode current mirror. The current, which is generated by the MOS operating in the subthreshold region, is nano-ampere magnitude [6].

The current-voltage (*I-V*) characteristic of MOSFETs in the subthreshold region is expressed as [7]:

$$I_D = K I_0 \exp(\frac{V_{GS} - V_{TH}}{\eta V_T})[1 - \exp(-\frac{V_{DS}}{V_T})] \tag{1}$$

$I_D$ is the drain-current, $K$ is the aspect ratio of the MOSFETs, $V_{GS}$ is the gate-source voltage, $V_{DS}$ is the drain-source voltage, $V_{TH}$ is the threshold voltage, $\eta$ is the sub-threshold slope factor, $I_0 = \mu C_{OX}(\eta - 1)V_T^2$ is characteristic current, in which $V_T = k_B T/q$ is the thermal voltage, $k_B$ is the Boltzmann constant, $q$ is the elementary charge, $\mu = \mu_0(T_0/T)^m$ is the carrier mobility, in which $\mu_0$ is the electron migration

rate of MOSFETs at room temperature, $T_0$ is reference temperature, $T$ is the absolute temperature, and $m$ is the mobility temperature exponent [8,9].

When $V_{DS} > 4V_T$, the effect of $V_{DS}$ can be ignored, and Equation (1) is simplified as Equation (2).

$$I_D = KI_0 \exp(\frac{V_{GS} - V_{TH}}{\eta V_T}) \tag{2}$$

$V_{GS}$ in the subthreshold region can be expressed as Equation (3).

$$V_{GS} = V_{TH} + \eta V_T \ln(\frac{I_D}{KI_0}) \tag{3}$$

In Figure 1, the current source circuit current $I_D$ is obtained by R1, M11 and M12, and can be expressed as Equation (4).

$$
\begin{aligned}
I_D &= \frac{V_{GSM11}(T) - V_{GSM12}(T)}{R_1} \\
&= \frac{V_{TH} + \eta V_T \ln(\frac{I_D}{K_{11}I_0}) - V_{TH} - \eta V_T \ln(\frac{I_D}{K_{12}I_0})}{R_1} \\
&= \frac{\eta V_T}{R_1} \ln \frac{K_{12}}{K_{11}} \\
&= \frac{\eta k_B T}{R_1 q} \ln \frac{K_{12}}{K_{11}}
\end{aligned}
\tag{4}
$$

From the above equation, the relationship between the current source circuit current $I_D$ and the temperature can be expressed as Equation (5).

$$\frac{\partial I_D}{\partial T} = \frac{\eta k_B}{R_1 q} \ln \frac{K_{12}}{K_{11}} \tag{5}$$

Current $I_D$ is copied from current source circuit to voltage reference circuit by setting a proper aspect ratio of the current mirror; a bias current is provided for M13–M20 in voltage reference circuit.

## 2.3. The Vref1 Generator Circuit

The first voltage reference *Vref1* is shown in Figure 1; the formula of the output *Vref1* can be obtained as Equation (6) [10].

$$
\begin{aligned}
V\text{ref1} &= V_{GS20} = V_{TH20} + \eta V_T \ln(\frac{aI_D}{K_{20}I_0}) \\
&= V_{TH0} - \kappa T + \eta \frac{k_B T}{q} \ln(\frac{a\eta V_T \ln \frac{K_{12}}{K_{11}}}{K_{20}\mu_0(T_0/T)^m C_{OX}(\eta-1)V_T^2 R_1}) \\
&= V_{TH0} - \kappa T + \eta \frac{k_B T}{q} \ln(\frac{a\eta \ln \frac{K_{12}}{K_{11}}}{K_{20}\mu_0(T_0/T)^m C_{OX}(\eta-1)V_T R_1})
\end{aligned}
\tag{6}
$$

$V_{TH} = V_{TH0} - kT$ is the threshold voltage. When $m = 0$, we can obtain the relationship between the first output reference voltage *Vref1* and the temperature as Equation (7).

$$\frac{\partial V\text{ref1}}{\partial T} = -\kappa + \eta \frac{k_B}{q} \ln(\frac{qa\eta \ln \frac{K_{12}}{K_{11}}}{K_{20}\mu_0 C_{OX}(\eta-1)k_B R_1}) \tag{7}$$

Solving for $dVref1/dT = 0$, the condition of zero temperature drift of the output voltage can be obtained as Equation (8).

$$\frac{\ln \frac{K_{12}}{K_{11}}}{K_{20}} = \frac{\exp(\frac{q\kappa}{\eta k_B})\mu_0 C_{OX}(\eta-1)k_B R_1}{qa\eta} \tag{8}$$

From Equation (8), we can see that the first output reference voltage *Vref1* with low temperature coefficient can be produced, with proper adjustment of $K_{11}$, $K_{12}$, $K_{20}$ and $R_1$.

## 2.4. The Vref2 Generator Circuit

The second voltage reference *Vref2* is shown in Figure 1. Low Temperature coefficient of *Vref2* is generated by the difference voltage between the gate-source voltages of 3.3-V MOS transistor M17 and 1.8-V MOS transistor M18, and overdrive voltages of 1.8-V MOS transistor $M_{13}$, $M_{14}$, $M_{15}$, and $M_{16}$. The output *Vref2* can be obtained as Equation (9) [4].

$$
\begin{aligned}
V\text{ref2} &= V_{\text{OV}13} + V_{\text{OV}15} - V_{\text{OV}14} - V_{\text{OV}16} + V_{\text{GSM}17} - V_{\text{GSM}18} \\
&= V_{\text{TH}17} - V_{\text{TH}18} + \eta V_{\text{T}} \ln\left(\frac{K_{14}K_{16}K_{18}(t_{\text{OX}14}t_{\text{OX}16}t_{\text{OX}18})}{K_{13}K_{15}K_{17}(t_{\text{OX}13}t_{\text{OX}15}t_{\text{OX}17})}\right)
\end{aligned}
\tag{9}
$$

$$
V\text{ref2} = \Delta V_{\text{TH}} + \eta V_{\text{T}} \ln\left(\frac{K_{14}K_{16}K_{18}(t_{\text{OX}14}t_{\text{OX}16}t_{\text{OX}18})}{K_{13}K_{15}K_{17}(t_{\text{OX}13}t_{\text{OX}15}t_{\text{OX}17})}\right)
\tag{10}
$$

$\Delta V_{TH} = V_{TH17} - V_{TH18}$ is the difference of $V_{TH}$ with negative temperature coefficient, $V_T = k_B T/q$ is the thermal voltage with positive temperature coefficient, so the voltage reference *Vref2* with low temperature coefficient can be produced. $V_{OVi}$ is the overdrive voltage of $M_i$. $t_{OX,i}$ is the oxide thickness of $M_i$. $\kappa = dV_{TH}/dT$ is the temperature coefficient of $V_{TH}$, as Equation (11), where $E_g$ is the bandgap, $\varepsilon_{Si}$ is the relatively dielectric constant of Si-substrate, $N_C$ is the effective density of states of conduction band, and $N_V$ is the effective density of states of valence band.

$$
\kappa = \frac{dV_{TH}}{dT} = -(2\eta - 1)\frac{k_B}{q}\left\{\ln\left(\frac{\sqrt{N_c N_v}}{N_A}\right) + \frac{3}{2}\right\} + \frac{\eta - 1}{q}\frac{dE_g}{dT}
\tag{11}
$$

$V_{TH}$ can be expressed as Equation (12).

$$
V_{TH} = -\frac{E_g}{2q} + \psi_B + \frac{\sqrt{4\varepsilon_{si}qN_A\psi_B}}{C_{OX}}
\tag{12}
$$

The relationship between the voltage reference *Vref2* and the temperature can be expressed as Equation (13).

$$
\frac{\partial V\text{ref2}}{\partial T} = \frac{\dfrac{(t_{OX14}t_{OX16}t_{OX18})-(t_{OX13}t_{OX15}t_{OX17})}{\varepsilon_{OX}}\sqrt{\dfrac{2N_A\varepsilon_{si}q^2}{2k_B T_0 \ln\left(N_A/\sqrt{N_c N_v}\right)+E_g}}}{\dfrac{k_B}{q}\ln\dfrac{N_A}{\sqrt{N_c N_v}} + \dfrac{\eta k_B}{q}\ln\left[\dfrac{K_{14}K_{16}K_{18}(t_{OX14}t_{OX16}t_{OX18})}{K_{13}K_{15}K_{17}(t_{OX13}t_{OX15}t_{OX17})}\right]}
\tag{13}
$$

Solving for d*Vref2*/d*T* = 0, the condition of zero temperature drift of the output voltage can be obtained as Equation (14).

$$
\frac{K_{13}K_{15}K_{17}}{K_{14}K_{16}K_{18}} = \frac{t_{OX13}t_{OX15}t_{OX17}}{t_{OX14}t_{OX16}t_{OX18}} \exp\left[\frac{\dfrac{(t_{OX14}t_{OX16}t_{OX18})-(t_{OX13}t_{OX15}t_{OX17})}{\eta\varepsilon_{OX}}}{\sqrt{\dfrac{2N_A\varepsilon_{si}q^2}{2k_B T_0 \ln\left(N_A/\sqrt{N_c N_v}\right)+E_g}}}\times\ln\frac{\sqrt{N_c N_v}}{N_A}\right]
\tag{14}
$$

With proper adjustment of $K_{13}$, $K_{14}$, $K_{15}$, $K_{16}$, $K_{17}$ and $K_{18}$, the voltage reference *Vref2* with low temperature coefficient can be produced.

Table 1 provides the dimensions of key components in the proposed circuit, including the aspect ratio of transistors and resistance value.

**Table 1.** Device size of the proposed circuit.

| Transistor | W/L (μm/μm) | Transistor | W/L (μm/μm) |
|---|---|---|---|
| M1 | 0.42/0.42 | M12 | 5.9/2 |
| M2 | 0.42/0.42 | M13 | 10/1 |
| M3 | 0.42/0.42 | M14 | 10/1 |
| M4 | 0.42/0.42 | M15 | 10/1 |
| M5 | 2*10/2 | M16 | 10/1 |
| M6 | 2*10/2 | M17 | 5.5/2 |
| M7 | 2*10/2 | M18 | 14/7 |
| M8 | 2*10/2 | M19 | 1.6/1.25 |
| M9 | 2.5/2 | M20 | 2.3/18 |
| M10 | 2.5/2 | **Resistance** | **(kΩ)** |
| M11 | 2.4/1 | R1 | 635.4 |

## 3. Simulation Results

The proposed dual-output voltage reference circuit was designed in TSMC 0.18-μm CMOS process. Figure 2 shows the layout of this circuit, which has an area of $362.6 \times 282.4$ μm$^2$, and an enlarged view of the core circuit, which takes an active area of $103.1 \times 84.1$ μm$^2$.

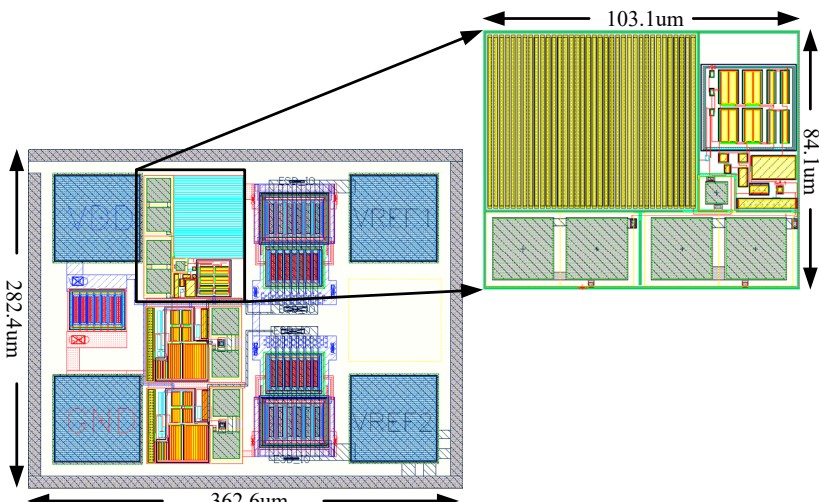

**Figure 2.** Layout of the dual-output voltage reference circuit.

To evaluate the performance of the proposed voltage reference and validate the design procedure, a series of simulations was carried out with the aid of SPICE simulator using TSMC 0.18-μm Mixed Signal/RF technology. Using the device's mismatch model, Monte Carlo simulation was run over a set of 2000 samples on a typical process corner and at room temperature. The results are illustrated in Figure 3. The mean values of the dual-output voltages *Vref1* and *Vref2* were 281.28 mV with a variation coefficient of 1.73% and 320.52 mV with a variation coefficient of 1.44%, respectively.

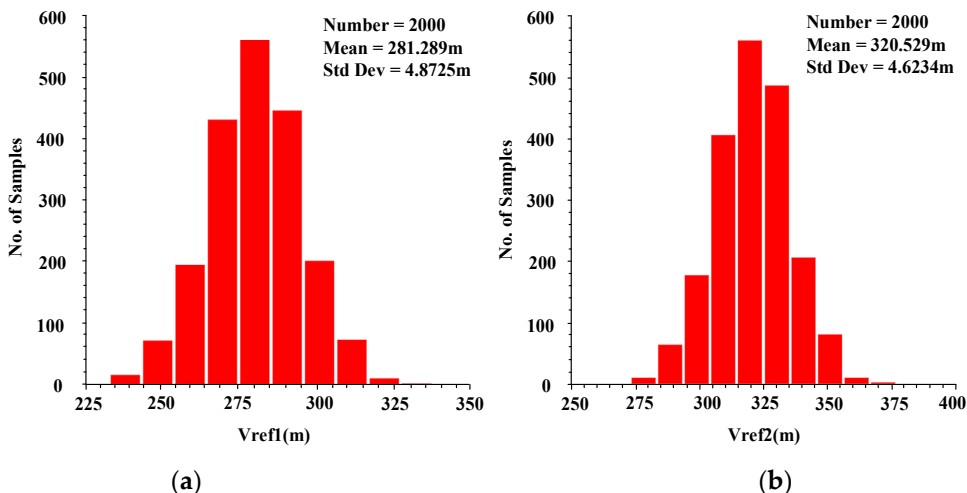

**Figure 3.** Monte Carlo simulation of output voltages for 2000 samples: (**a**) simulation *Vref1*; and (**b**) simulation *Vref2*.

A series of simulations was done to verify the performance of the proposed voltage reference. Figures 4 and 5 show the simulation results in different process corners.

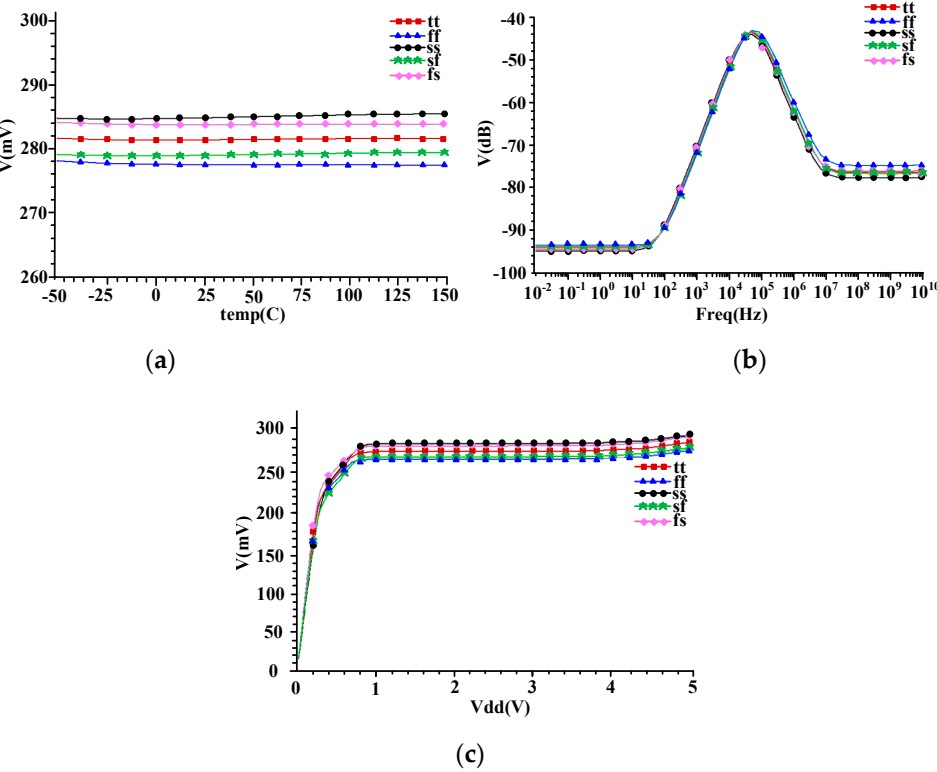

**Figure 4.** Simulation of *Vref1* on different process corners: (**a**) simulation TC of *Vref1*; (**b**) simulation PSRR of *Vref1*; and (**c**) simulation LS of *Vref1*.

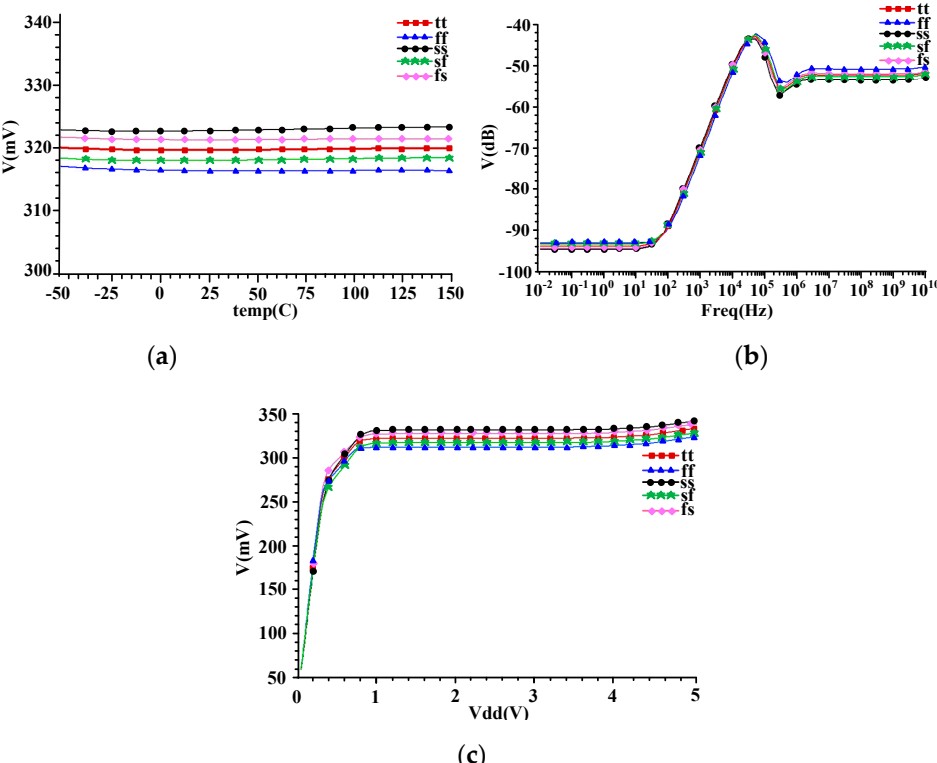

**Figure 5.** Simulation of *Vref2* on different process corners: (**a**) simulation TC of *Vref2*; (**b**) simulation PSRR of *Vref2*; and (**c**) simulation LS of *Vref2*.

## 4. Discussion

Table 2 summarizes the main characteristics of the proposed voltage reference on different process corners. In the temperature range from −40 to 150 °C, the TCs of output voltages were 15.2 ppm/°C and 16.8 ppm/°C, respectively. The mean LS was 0.11% under a supply voltage ranging from 0.9 V to 3.1 V at room temperature. The PSRR of *Vref1* at 100 Hz and 1 MHz was greater than −88.9 dB and −89.9 dB, respectively. The PSRR of *Vref2* at 100 Hz and 1M Hz was greater than −52.8 dB and −76.7 dB, respectively. The minimum power consumption was only 84.1 nW at 0.9 V supply power.

**Table 2.** The characteristics of the proposed voltage reference on different process corners.

| Corner | TC (−40°C–150°C) (ppm/°C) | PSRR (dB) | | LS (0.9 V–3.1 V) (%/V) | Min Power (nW) |
|---|---|---|---|---|---|
| | | @100 Hz | @1 MHz | | |
| *Vref2*tt | 16.8 | −88.9 | −52.8 | 0.11 | 81.4 |
| *Vref1*tt | 15.2 | −89.9 | −76.7 | 0.11 | |
| *Vref2*ff | 31.2 | −89.4 | −51.4 | 0.10 | 94.05 |
| *Vref1*ff | 28.5 | −90.2 | −75.1 | 0.10 | |
| *Vref2*ss | 36.5 | −88.6 | −54.1 | 0.08 | 73.3 |
| *Vref1*ss | 42.7 | −89.6 | −78.1 | 0.08 | |
| *Vref2*fs | 18.1 | −89.0 | −52.5 | 0.08 | 77.5 |
| *Vref1*fs | 12.9 | −89.8 | −77.0 | 0.08 | |
| *Vref2*sf | 26.1 | −88.8 | −53.2 | 0.09 | 85.45 |
| *Vref1*sf | 29.1 | −89.8 | −76.4 | 0.09 | |

In Table 3, the main performance of the proposed voltage reference in this circuit are summarized and compared with other published voltage references. It can be noticed that the proposed reference has many advantages, such as lower temperature coefficient, higher power supply rejection ratio, lower power and occupied the smallest layout area.

**Table 3.** The performance comparison of proposed voltage reference.

| Parameter | | [1] | [2] | [3] | [11] | [12] | This Work |
|---|---|---|---|---|---|---|---|
| CMOS technology (nm) | | 180 | 180 | 180 | 130 | 180 | 180 |
| Supply voltage (V) | | 0.45–1.8 | 1.3–1.8 | 1.1–1.8/0.7–1.8 | 1.2 | 0.85–2.5 | 0.9–3.1 |
| VREF (mV) | | 118.46 | 547 | 1090/548 | 735 | 634.1 | 320.5/281 |
| Temperature Range (/°C) | | −40–125 | −40–140 | −40–120 | −40–120 | −20–80 | −40–150 |
| Average TC (ppm/°C) | | 63.6 | 1.67 | 147/114 | 9.3 | 16.3 | 16.8/15.2 |
| Line Sensitivity (%/V) | | 1.2 | 0.08 | 0.737/1.05 | - | 0.086 | 0.11/0.11 |
| PSRR (dB) | @100 Hz | −44.2 | −55 | −62/−56 | −30 | −83 | −88.9/−89.9 |
| | @1 MHz | −26.3 | - | - | - | >−64 | −52.8/−76.7 |
| Min Power (nW) | | 14.4@4.5 V | - | 100@1.1/52.5@0.7 V | 30.5@0.6 V | 202@0.85 V | 81.4@0.9 V |
| Area (mm$^2$) | | 0.012 | 0.0094 | 0.054 | 0.063 | 0.01 | 0.0086 |

## 5. Conclusions

The proposed circuit was designed in a standard 0.18-μm CMOS process. The simulation results show that, at temperature ranging from −40 to 150 °C, the TCs of output voltages were 15.2 ppm/°C and 16.8 ppm/°C, respectively. The mean LS was 0.11% under a supply voltage ranging from 0.9 V to 3.1 V at room temperature. The PSRRs of *Vref1* and *Vref2* at 100 Hz and 1 MHz were greater than −88.9 dB and −89.9 dB, and −52.8 dB and −76.7 dB, respectively. The results of 2000-point Monte Carlo simulation show that the output voltages 281 mV and 320.5 mV had variation coefficients of 1.73% and 1.44%, respectively. The minimum power consumption was 84.1 nW at 0.9 V supply, and the layout area of the proposed voltage reference was 0.0086 mm$^2$. It was more suitable for low-power fields, such as human body area networks, wearable medical devices and medical measurements, especially portable biomedical application.

**Author Contributions:** Conceptualization, X.S.; Formal analysis, X.S. and J.L.; Funding acquisition, H.Y.; Investigation, H.Y., T.W. and S.L.; Methodology, X.S. and J.L.; Project administration, H.L.; Resources, H.Y.; Supervision, W.X. and H.L.; Visualization, T.W. and S.L.; Writing—original draft, X.S. and J.L.; and Writing—review and editing, W.X., H.L. and B.W.

**Funding:** This research was funded by the National Natural Science Foundation of China (Grant Nos. 61861009 and 11264009), the Guangxi Natural Science Foundation (No. 2017GXNSFAA198224), GUET Excellent Graduate Thesis Program (Nos. 16YJPYSS06 and 16YJPYSS07), and Innovation Project of GUET Graduate Education (Nos. 2017YJCX35 and 2017YJCX31).

**Conflicts of Interest:** The authors declare no conflict of interest.

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
