# Peer review of "16.8/15.2 ppm/°C 81 nW High PSRR Dual-Output Voltage Reference for Portable Biomedical Application"

_electronics, doi:10.3390/electronics8020213_

Round 1

Reviewer 1 Report

The manuscript describes a new method to improve the performance of the multiple voltage reference, with a low power dissipation, low TC and high PSRR circuit. The strong point of the paper is the proposed architecture which includes a circuit consisting of only one start-up circuit, only one current reference circuit, and two voltage reference circuits, as well as the the small area of the total circuit. In conclussions it would add value to the article if a list of potential applications of the proposed circuit is mentioned. The article in my opinion is suitable for publication

Author Response

Point 1: The manuscript describes a new method to improve the performance of the multiple voltage reference, with a low power dissipation, low TC and high PSRR circuit. The strong point of the paper is the proposed architecture which includes a circuit consisting of only one start-up circuit, only one current reference circuit, and two voltage reference circuits, as well as the small area of the total circuit. In conclusions it would add value to the article if a list of potential applications of the proposed circuit is mentioned. The article in my opinion is suitable for publication.

Response 1: Thank you for arranging a timely review for our manuscript, and thank you for your approval of this study. We have carefully evaluated the reviewers’ critical comments and thoughtful suggestions. In conclusions, a list of potential applications of the proposed circuit has been added in this paper

As:

It is more suitable for low-power fields, like human body area networks, wearable medical devices and medical measurements, especially portable biomedical application.

Reviewer 2 Report

This paper presents a dual-output voltage reference operating in subtreshold. The authors claim nanopower consumption with very good performance metrics, making the proposed voltage reference suitable for biomedical applications.

The article has very poor language (phrasing and grammar), must be  improved consistently!

The literature review in Section 1 is very good. 

The proposed circuit, figure 1, is developed around some classical structures which are not the original contribution of the authors: a startup circuit (reference needed), a beta-multiplier self-biasing circuit for the generation of the bias current (reference needed), the Vref1 generator which is also provided in literature (e.g. Yunling Luo et al, 2012, DOI:10.1109/EDSSC.2012.6482788), and the Vref2 generator which is provided in reference 4.

The only original contribution of the authors is the dual output, accordingly, the proposed circuit generates two different reference voltages as a result of two strategies.

The gate-source voltage difference to achieve low temperature coefficient for Vref2 is not well explained. Equation 9 doesn't stand. M17-M15-M16-M18 is not a translinear loop (at least not as illustrated in Fig.1), and neither does the equation stand as Kirchoff's law. 

As explained in reference 4, the bias current a*ID of the Vref2 circuit does have a contribution on the reference voltage. According to equation 14 in this article, it does not.     

Equations 8 and 14 specify the design equations for the transistor aspect ratios and the value of the passive resistance. Please specify some numeric values. Also, what is teh order o magnitude of the passive resistance, is it feasible to be integrated as illustrated in Fig. 2?

Provide more details about the simulation procedure: what simulation environment was used, simulation parameters, etc. 

Author Response

Point 1: This paper presents a dual-output voltage reference operating in subthreshold. The authors claim nanopower consumption with very good performance metrics, making the proposed voltage reference suitable for biomedical applications.

The article has very poor language (phrasing and grammar), must be improved consistently!

The literature review in Section 1 is very good.

Response 1: Thank you for arranging a timely review for our manuscript, and thank you for your approval of this study. We have carefully evaluated the reviewers’ critical comments and thoughtful suggestions. In addition, we have corrected some grammatical errors, and revised some sentence patterns and phrasings.

Point 2: The proposed circuit, figure 1, is developed around some classical structures which are not the original contribution of the authors: a start-up circuit (reference needed), a beta-multiplier self-biasing circuit for the generation of the bias current (reference needed), the Vref1 generator which is also provided in literature (e.g. Yunling Luo et al, 2012, DOI:10.1109/EDSSC.2012.6482788), and the Vref2 generator which is provided in reference 4.

Response 2: Thanks, we have revised the Reference to make them clearer, and add literature [10] (Yunling Luo et al, 2012, DOI:10.1109/EDSSC.2012.6482788) in the Reference. And add literature [4] in 2.4. The Vref2 generator circuit.

Point 3: The gate-source voltage difference to achieve low temperature coefficient for Vref2 is not well explained. Equation 9 doesn't stand. M17-M15-M16-M18 is not a translinear loop (at least not as illustrated in Fig.1), and neither does the equation stand as Kirchoff's law.

As explained in reference 4, the bias current a*ID of the Vref2 circuit does have a contribution on the reference voltage. According to equation 14 in this article, it does not.

Response 3: Thanks, we have revised 2.4. The Vref2 generator circuit to explain low temperature coefficient for Vref2. We think it is necessary to rewrite formula (9)-(14) about theoretical derivation of the TC of Vref2.

As:

Low Temperature coefficient of Vref2 is generated by the difference voltage between the gate-source voltages of 3.3-V MOS transistor M17 and 1.8-V MOS transistor M18, and overdrive voltages of 1.8-V MOS transistor M13, M14, M15, M16. The output Vref2 can be obtained as formula (9)[4].

                                                                                  (9)

                                  (10)

ΔVTH=VTH17-VTH18 is the difference of VTH with negative temperature coefficient, VT= kBT/q is the thermal voltage with positive temperature coefficient, so the voltage reference Vref2 with low temperature coefficient can be produced. VOVi is the overdrive voltage of Mi. tOX,i is the oxide thickness of Mi. = dVTH/dT is the temperature coefficient of VTH, as formula (11), where Eg is the bandgap, εSi is the relatively dielectric constant of Si-substrate, NC is the effective density of states of conduction band, NV is the effective density of states of valence band.

                                       (11)

VTH can be expressed as formula (12).

                                                  (12)

The relationship between the voltage reference Vref2 and the temperature can be expressed as formula (13).

                       (13)

Solving for dVref2/dT=0 , then the condition of zero temperature drift of the output voltage can be obtained as formula (14).

                   (14)

With proper adjustment of K13, K14, K15, K16, K17 and K18, the voltage reference Vref2 with low temperature coefficient can be produced.

Point 4: Equations 8 and 14 specify the design equations for the transistor aspect ratios and the value of the passive resistance. Please specify some numeric values. Also, what is teh order o magnitude of the passive resistance, is it feasible to be integrated as illustrated in Fig. 2?

Response 4: Thanks, we have added the Table 1. The sizes of the key components in the proposed circuit are listed in the Table 1, all the aspect ratio of transistors and resistance value. The passive resistance is 635.4kΩ, it possible to be integrated.

As:

Table 1 provides the dimensions of key components in the proposed circuit, including the aspect ratio of transistors and resistance value.

Table 1. Device size of the proposed circuit.

Transistor

W/L(um/um)

Transistor

W/L(um/um)

M1

0.42/0.42

M12

5.9/2

M2

0.42/0.42

M13

10/1

M3

0.42/0.42

M14

10/1

M4

0.42/0.42

M15

10/1

M5

2*10/2

M16

10/1

M6

2*10/2

M17

5.5/2

M7

2*10/2

M18

14/7

M8

2*10/2

M19

1.6/1.25

M9

2.5/2

M20

2.3/18

M10

2.5/2

Resistance

(kΩ)

M11

2.4/1

R1

635.4

Point 5: Provide more details about the simulation procedure: what simulation environment was used, simulation parameters, etc.

Response 5: Thanks, we have added the simulation procedure, like simulation environment, simulation parameters, etc.

As:

In order to evaluate the performance of the proposed voltage reference and validate the design procedure, a series of simulations are carried out with the aid of SPICE simulator using TSMC 0.18-μm Mixed Signal/RF technology. Using the device’s mismatch model, Monte Carlo simulation has been run over a set of 2000 samples on the typical process corner and room temperature.

Round 2

Reviewer 2 Report

I consider that all my remarks have been addressed and the points I have raised were consistently improved. Therefore, I am confident to accept this paper for publication.